# Towards Sustainable Composting of Source-Separated Biodegradable Municipal Solid Waste—Insights from Long An Province, Vietnam

**Tan Loi Huynh [1]** , **Thi Kim Oanh Le [1]** , **Yong Jie Wong [2],\*** , **Chi Tuong Phan [1]** and **Thi Long Trinh [3]**

1   Faculty of Environment, School of Technology, Van Lang University, Ho Chi Minh City 70000, Vietnam; loi.ht@vlu.edu.vn (T.L.H.); oanh.ltk@vlu.edu.vn (T.K.O.L.); tuong.pc@vlu.edu.vn (C.T.P.)
2   Department of Bioenvironmental Design, Faculty of Bioenvironmental Sciences, Kyoto University of Advanced Science, Kameoka City 606-8501, Japan
3   World Wide Fund—Vietnam, Ho Chi Minh City 70000, Vietnam; long.trinhthi@wwf.org.vn
\*   Correspondence: wong.yongjie@kuas.ac.jp

**Abstract:** Inadequate municipal solid waste (MSW) management has become a pressing concern, resulting in significant environmental contamination, particularly in developing countries. Composting has demonstrated its practicality and feasibility for addressing this issue; however, the lack of at-source solid waste separation remains a major challenge. As a result, in this study, the first sustainable MSW separation at source was conducted in Tan An City, Long An Province. The objective of this study was to evaluate the compost process and quality using Tan An City's separated biodegradable organic solid waste as the raw material, through a windrow composting process with active aeration. Biodegradable organic waste, slow-biodegradable organic waste and plastic waste accounted for 84.5%, 15.1% and 0.4%, respectively, of the total waste. The pH, moisture, volatile solid percentage, total nitrogen, total organic carbon and carbon-to-nitrogen ratio of the separated solid waste were $8.7 \pm 0.4$, $76.8 \pm 1.9\%$, $68.3 \pm 1.3\%$, $2.1 \pm 0.1\%$, $35.7 \pm 2.2\%$ and $17 \pm 0.8$, respectively. Rice straw was mixed with solid waste as a bulking material in a 31%:69% ratio to achieve a moisture content of 55% in the mixture. After 10 weeks, an evaluation of the compost's quality revealed its potential suitability for agricultural applications. Notably, Salmonella was not detected in the compost, and the heavy metal levels were below standard limits, indicating the safety of the compost. To ensure optimal nutrient levels for effective plant growth, a slight nitrogen and phosphorus supplement was recommended. Aligned with the C/N ratio of 12.1 and a consistent temperature of approximately 29 °C, this indicates a high degree of maturity and stability in the composting process. The framework of this study demonstrates the effectiveness of at-source MSW separation in paving a sustainable path for MSW management.

**Keywords:** source-separated biodegradable municipal solid waste; windrow compost; nutrients recovery; waste management



## 1. Introduction

Sustainable municipal solid waste (MSW) management has remained a significant and persistent environmental challenge globally, particularly in developing and low-income countries. According to [1], the MSW generation has reached alarming levels, with approximately 2.24 billion tonnes produced in 2020, equivalent to an average of 0.79 kg of waste per capita per day. According to projections, MSW generation will increase by 73% by 2050 compared to 2020 levels, reaching a staggering 3.88 billion tonnes, corresponding to an average of approximately 1.09 kg of waste per capita per day, which could be attributed to booming populations, uncontrolled urbanisation and rapid economic development [2]. Improper MSW management can lead to severe environmental consequences, including groundwater contamination, climate change and ecosystem degradation [3,4].

Along with the current need to reduce environmental risk, the United Nation Sustainable Developments Goals (UNSDGs) introduced in 2015 provide a blueprint for managing global challenges. SDG 12 specifically promotes responsible consumption and production by emphasizing the need to implement waste prevention, waste reduction, recycling and reuse strategies to reduce the environmental impacts associated with MSW generation. As a result, the global transition to a circular economy has accelerated in the pursuit of sustainable MSW management. At present, various frameworks/collaborations for sustainable MSW management have been developed, such as the Global Alliance for Incinerator Alternatives, the Zero Waste International Alliance, the Waste and Resources Action Programme, the Alliance to End Plastic Waste, the Basel Convention and the Advancing Sustainable Materials Management by United States Environmental Protection Agency [5]. However, a lack of consistent financial support, transparent policy implementation and community engagement hinder the implementation progress in many regions, particularly in developing Southeast Asian countries. For instance, the governments of Cambodia and Indonesia seek financial support from international organizations, such as the United Nations Environment Program and the Japan International Cooperation Agency for technical assistance in waste management and disposal, as the reported coverage of their populations with SWM is only between 21% and 40% [6]. Therefore, this underscores the need for localized solutions that combine innovative technology, effective policy implementation, sustainable financial capacities and robust community engagement for archiving sustainable MSW management.

The Ministry of Natural Resources and Environment (MONRE) in Vietnam estimates that the generation rate of municipal solid waste increases by 10% to 16% each year [7]. In most cities, municipal solid waste accounted for 60 to 70% of total domestic solid waste. To address this issue, the Vietnamese government issued the Law on Environmental Protection 2020, which took effect on 1 January 2022; however, solid waste separation at the source has yet to be successfully implemented in any Vietnamese city [8]. As a result of these and with the assistance of the World Wide Fund for Nature in Vietnam (WWF-Vietnam), the first solid waste separation at the source was conducted in Tan An City, Long An Province.

The collected municipal solid waste is currently treated by landfill disposal and incineration. Long An Province's solid waste is treated at composting plants in an unspecified proportion. However, the compost product does not meet the agricultural application standard because of the lack of nutrients, size of the particles and impurity with plastic, metal and other inorganic wastes. The application of recycling, reuse and recovery of resources from waste has the potential to help Long An Province improve its environmental condition as it moves toward a circular economy and sustainable development.

Combining solid waste composting with circular economy principles can help address waste generation's environmental and social challenges. The circular economy promotes waste reduction and resource efficiency by reusing materials for as long as possible, extracting the maximum value from them and then recovering and regenerating them at the end of their life. In the case of the Long An Province, integrating circular economy principles into waste management could enhance compost quality, decrease the reliance on landfills and incineration, lower greenhouse gas emissions and foster local economic growth through innovative recycling and reuse practices. Composting solid waste constitutes a crucial element of a circular economic waste management approach, enhancing sustainability and resilience by recuperating value from waste and mitigating greenhouse gas emissions. Consequently, this research employs a case study to demonstrate the quality of compost derived from segregated solid waste at its source, while also evaluating the viability of adopting solid waste composting as part of Vietnam's transition towards a circular economy.

## 2. Materials and Methods

### 2.1. Study Area

Tan An City, the capital of Long An Province, is a flourishing urban centre, strategically located within the Mekong Delta region and in proximity to Ho Chi Minh City. With a population of around 200,000 people, the city thrives within an area spanning 81.94 km$^2$ (Figure 1). Tan An City geographically benefits from its placement along the Mekong Delta, a fertile region known for its agricultural productivity that contributes to the city's economic prowess, making it a vital component in the province's agricultural output [9]. The city's per capita income stands at approximately USD 2400 per annum. Tan An City has also been designated as an Asian Development Bank pilot demonstration site for proper waste management from December 2019 to July 2023, demonstrating its commitment to environmental sustainability [10]. Anticipating the adoption of solid waste separation at the source, Tan An City partnered with WWF-Vietnam to equip households within the study area with three waste bins each. A total of 10,000 bins with a capacity of 28 L, similar to the common size of household waste receptacles in Vietnam, were distributed to the Tan An City residents [11–13]. Waste management issues are being addressed through the implementation of waste separation and recycling programmes. These initiatives aim to minimise the environmental impact and promote responsible waste disposal practices, which are in line with global sustainability goals.

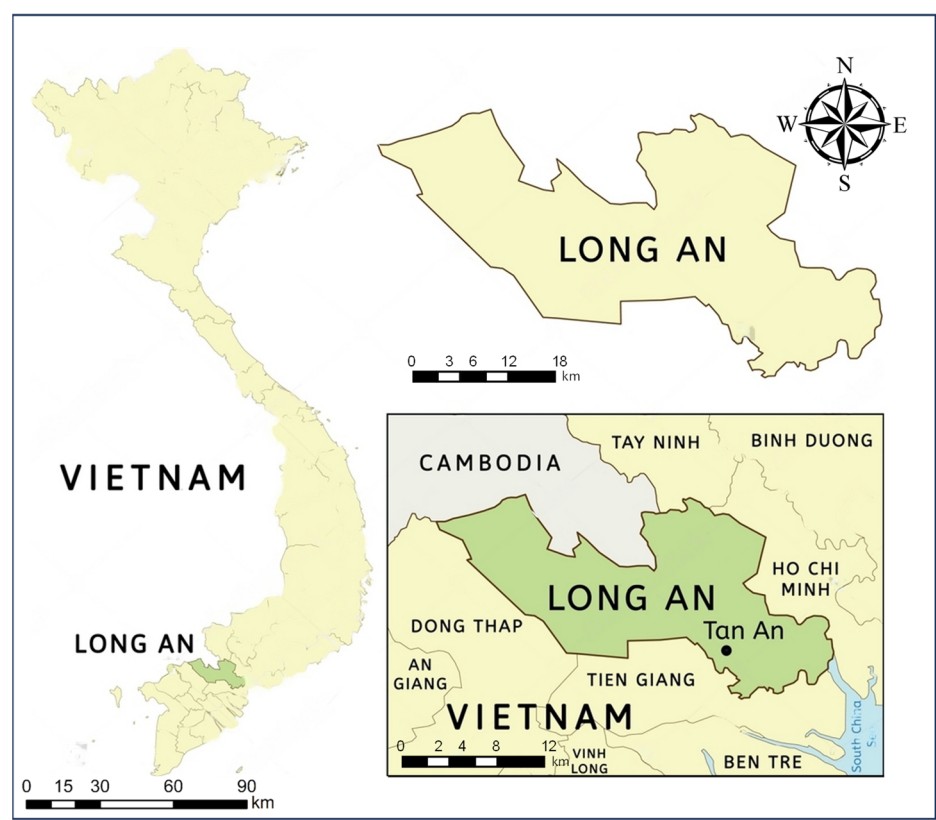

**Figure 1.** Geographical location of Tan An City, Long An Province, Vietnam. Adapted with permission from [14].

### 2.2. Materials

Organic solid waste was collected in Ward 3 of Tan An City, Long An Province, where a source separation system has been in place since 2021. The 2.1 tonnes of separated organic waste were then carefully transported to the ambient laboratory of Van Lang University, Campus 3, ensuring the preservation of its inherent characteristics. Prior to any further processing, it was imperative to conduct a thorough quality check on the solid waste following its separation at the source. The quality assessment is a valuable indicator of the

efficacy of the waste separation practices used. Upon the second separation conducted for separation assessment, 1.8 tonnes of separated organic solid waste from Ward 3, Tan An City, was shredded before being collected in representative samples using the quartering method. To obtain a comprehensive understanding of the waste composition, five samples (500 g/sample) were chosen at random and analysed across five essential parameters: pH, moisture content, volatile solids (VS), total organic carbon (TOC) and total nitrogen (T-N). These analytical results play a pivotal role in determining the composition of the organic matter, which serves as the primary input material for the subsequent composting process. Along with organic matter determination, the pH level, carbon-to-nitrogen (C/N) ratio and moisture content are key factors that significantly influence both the composting process itself and the quality of the final compost product.

*2.3. Active Aeration Composting*

The implementation of the windrow with active aeration composting process is used in this study. This process offers two key advantages: (i) it utilises control systems in forced aeration to effectively regulate the temperature and (ii) it allows for the treatment of exhaust gases to control unpleasant odours.

The composting material used in this study consists of the organic fraction of solid waste obtained from Tan An City, along with a bulking material. Based on their respective properties, the specific mixing ratio between the organic fraction of solid waste and the bulking material is determined. After that, 1.8 tonnes of shredded waste with rice straw were mixed then divided into two windrows and composted for 10 weeks [15,16]. Each windrow has a dimension of $0.5 \times 0.5 \times 2.5$ m (height × width × length). To ensure an adequate oxygen supply for the microorganisms involved in the composting process, an aerated static pile system is designed. Individual piles are spaced 0.5 m apart in this system, with aeration pipe holes evenly distributed at 20 cm intervals. The holes are drilled at a 45-degree angle with respect to the ground. An extended pile configuration is another option, in which piles are built adjacent to each other without physical separation. This configuration allows for the processing of a larger amount of material within a given area.

As shown in Figure 2, the pilot composting operation follows a defined procedure. The organic solid waste is initially received at the laboratory area and undergoes segregation to separate slow-biodegradable and non-biodegradable components. Subsequently, the waste is shredded to reduce its size. The composition of both the solid waste and bulking material is analysed to determine the appropriate mixing ratio, ensuring an optimum moisture content and C/N ratio for the composting process. The organic solid waste and bulking material mixture is then used to set up the two windrows [17].

The blower is activated after it has been set up, and the uniformity of the windrow along the air supply pile systems is monitored. Throughout the composting process, the windrow is regularly assessed for temperature, pH and moisture content, which are adjusted as necessary to maintain optimal conditions for aerobic digestion. Simultaneously, manual mixing is performed on a daily basis to ensure uniformity within the windrow. The windrows continue to operate until the odour dissipates and the temperature stabilised, fluctuating within the ambient temperature range [17]. The process is expected to be completed after a 10-week composting period. The desired characteristics of the resulting compost include an earthy smell (not putrid, acrid or foul), fine particles without any visible remnants of the original waste or large pieces and a dark brown appearance.

The compost product obtained from the windrow composting process with active aeration will undergo sampling and compositional analysis. This analysis aims to evaluate the product against 17 parameters in order to compare its quality with the national fertiliser standard, specifically the standard known as 10 TCN 526-2002 [18]. The 17 parameters to be assessed include pH, moisture content, compost stability, particle size, number of microorganisms, total organic carbon (TOC), potassium oxide ($K_2O$), phosphorus pentoxide ($P_2O_5$), total nitrogen (T-N), presence of Salmonella, zinc (Zn), copper (Cu), lead (Pb), cadmium (Cd), chromium (Cr), nickel (Ni) and mercury (Hg).

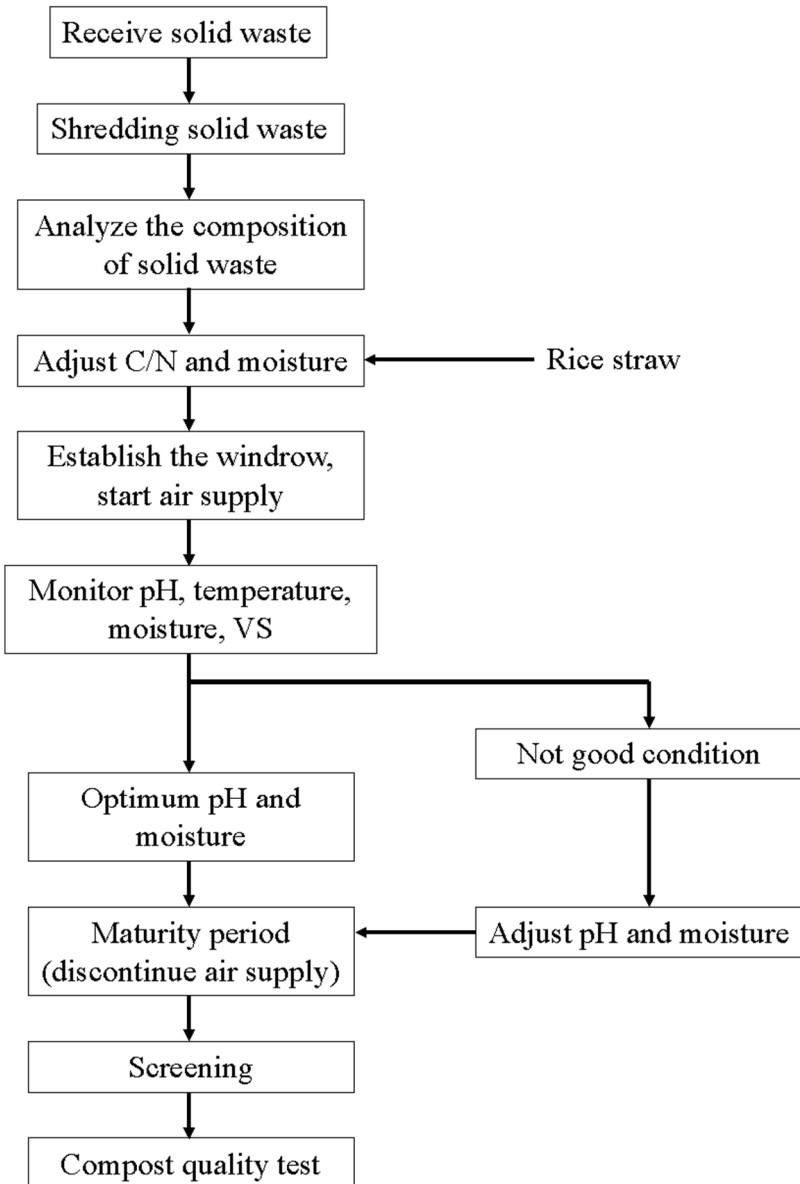

**Figure 2.** Composting experiment flow chart.

The pH was measured using a portable pH meter (Hach HQ11d). The moisture content of the compost was determined by heating 50.0 g of samples to a constant weight in a crucible placed in an oven (Binder ED53, Tuttlingen, Germany) maintained at 105 °C for 24 h. Compost stability, indicated by temperature variation after the composting process was completed, was determined using a thermometer. The percentage of particle sizes in the compost was assessed using 2 mm and 5 mm sieves. The number of microorganisms, as well as the presence of Salmonella, were determined using the plate count and streak plate method, respectively.

The concentrations of TOC and T-N were analysed using a TOC analyser and the Kjeldahl digestion method, respectively, while $P_2O_5$ levels were determined using the molybdovanadophosphate colorimetric method. For the remaining parameters, analysis was conducted using acid digestion combined with Inductively Coupled Plasma-Atomic Emission Spectrometry (ICP-AES).

To conduct the analysis, two samples per windrow will be collected and subjected to comprehensive testing. The compost product's composition will be compared to the specified requirements outlined in the national fertiliser standard 10 TCN 526-2002. This

comparative analysis will determine whether the compost meets the necessary criteria for it to be considered standard-compliant.

## 3. Results

### 3.1. Solid Waste Management in Tan An City, Long An Province

Currently, Long An Province generates 1086 tons of MSW per day, with a solid waste generation per capita of 0.64 kg/person/day [19]. The daily amount of municipal solid waste generated in Tan An city is approximately 130 to 150 tons, with the collection and treatment costs approximately 130 to 150 million dong. Prior to this study, the MSW collected from this area was transported to the Tam Sinh Nghia treatment plant, where it was disposed using composting and incineration processes [20]. However, the composted product does not meet the fertiliser standards. Additionally, the heat generated during the incineration process is not being utilised for other purposes. Furthermore, the exhausted air from the incineration process may contribute to the air pollution problem [21].

In this study, the MSW separation at the source began in July 2020, thanks to the efforts of the Vietnamese government and WWF-Vietnam. Waste separation bins were provided by the government for recyclables (paper, cardboard, metals, etc.), organic and residual wastes (including sanitary pads, diapers, toilet paper, packaging, rubber, broken glass/ceramic, cloth, yarn, towels, cutlery, etc.) (Figure 3a). Through raising awareness and effective policy implementation, the residents are committed to separating their MSW into categories (Figure 3b). Prior to collecting organic waste for composting, the garbage collector will thoroughly inspect the waste to ensure only organic waste is included in the next procedure.

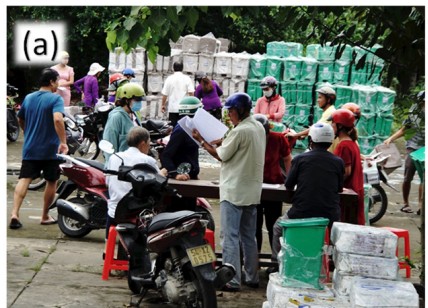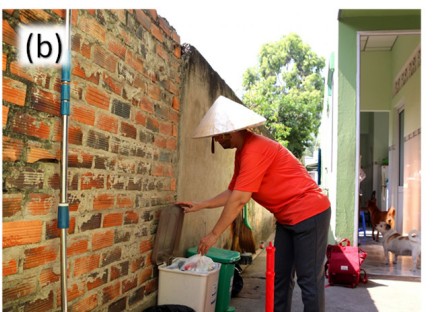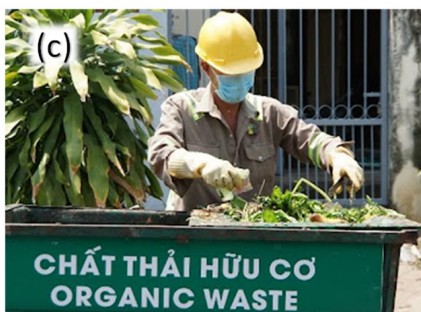

**Figure 3.** Municipal solid waste separation in Tan An City. (**a**) Residents receiving waste separation bins from the government; (**b**) residents placing these bins in front of their houses, demonstrating their commitment to proper waste disposal; and (**c**) garbage collector is thoroughly inspecting and separating residual waste, ensuring the solid waste is appropriately separated. Adapted with permission from [11–13].

In 2022, the MSW collected from households was evaluated, and the composition after separation is shown in the Figure 4. The recyclable waste is collected and delivered to recycling facilities. Following collection, the biodegradable organic waste is transferred to a composting plant, while the remaining waste is disposed of at the sanitary landfill. The results in Figure 4 show that the potential for recycling and resource recovery of recyclable and biodegradable organic waste accounted for 47.2% of the MSW generated. If source separation is extended to all of Tan An City, the amount of recyclable solid waste might reach 61.4–70.8 tons/day, which is relevant to the amount of solid waste reduction in landfills.

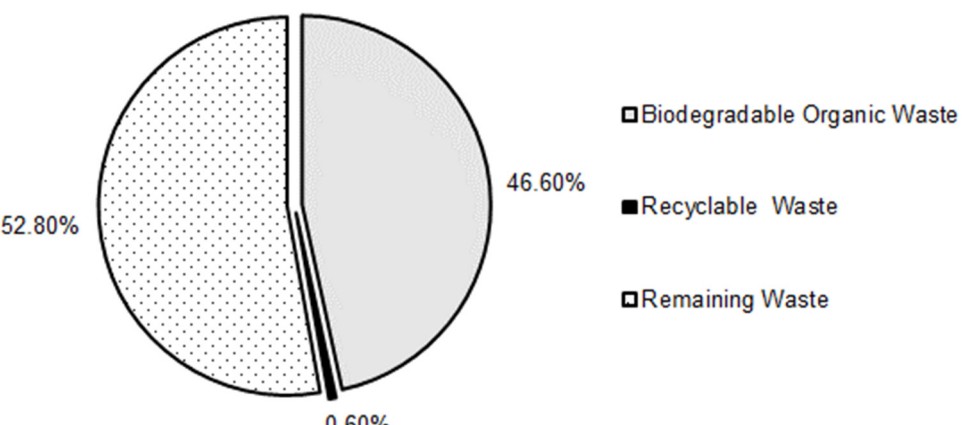

**Figure 4.** Waste percentage composition of MSW collected from household waste separation bins.

*3.2. Quality Checking of the Separated Solid Waste*

Within two days, the separated solid waste was collected from Ward 3, Tan An City, Long An Province, and transported to Van Lang University. The quality of the separated solid waste (2.1 tons) sent to Van Lang University Campus 3 was checked, and non-biodegradable organic matter was removed. The solid waste was classified into four groups: (1) coconuts and durian shells; (2) garden waste; (3) plastic, including plastic bottles and plastic bags; and (4) organic waste. Figure 5 shows the proportions of these groupings. Biodegradable organic waste, slow-biodegradable organic waste and plastic waste were accounted for 84.5%, 15.1% and 0.4%, respectively. The slow-biodegradable organic matter includes coconut and durian shells and tree branches, which take months or even a year to decompose [22]. Non-biodegradable plastic products, such as bottles and plastic bags, account for 0.4% of the total. Although it only accounts for a minor portion of the volume, its frequency is high. In order to improve the separation step prior to composting, the solid waste should not include plastic waste.

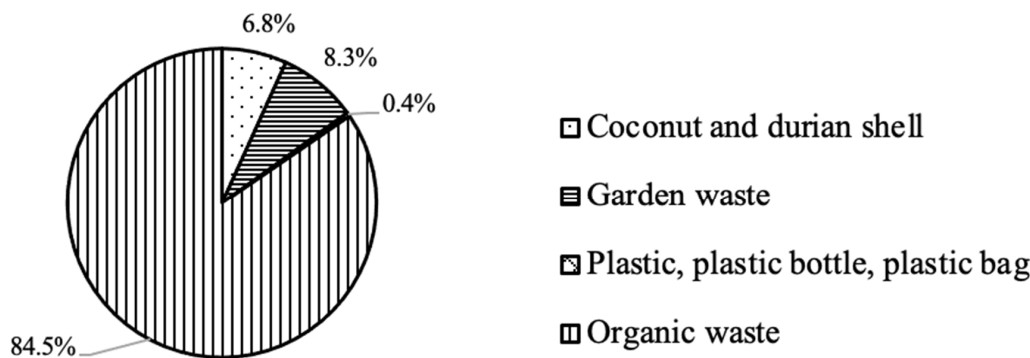

**Figure 5.** Composition of separated solid waste from Tan An City.

In this study, only biodegradable organic waste was used as an ingredient in the composting process, removing slow and non-biodegradable components including coconut and durian shells, garden waste and plastic waste. For slow-biodegradable organic ingredients that are not included in the input materials due to their long decomposition time of several months to a year, there is a possibility of prolonging the decomposition process. Furthermore, the mixture of coconut shells, durian shells and garden waste can be used as a mixing material to create porosity and provide a carbon source for the compost windrow. However, due to the limitations of the shredder equipment, this component was omitted in this study.

### 3.3. Characteristic of Separated Solid Waste

The separated solid waste was shredded, and the composition was analysed, as shown in Table 1. The result indicated that the pH of solid waste from Tan An city is slightly higher than the neutral range (pH = 8.7). The moisture content is 76.8%, which is similar to the findings of previous studies [23]. The C/N ratio of the solid waste is a bit low (17 ± 0.8). Therefore, mixing with base material is recommended to adjust the pH, moisture content and C/N ratio. Rice straw was chosen as the bulking material in this study. The characteristics of rice straw are shown in Table 1. The rice straw used in this study was collected from Hoc Mon District, Ho Chi Minh City.

**Table 1.** Characteristic of the separated solid waste and rice straw.

| No. | Parameter | Unit | This Study (*n* = 5) | Rice Straw (*n* = 1) | Previous Study [13] | Optimal Conditions [17] |
|---|---|---|---|---|---|---|
| 1 | pH | - | 8.7 ± 0.4 | 7.2 | 5.9 ± 0.3 | 6.0–7.5 |
| 2 | Moisture | % | 76.8 ± 1.6 | 6.5 | 80.9 ± 3 | 50–60 |
| 3 | VS | % | 68.3 ± 1.1 | 90.5 | 86.2 ± 3.8 | - |
| 4 | T-N | % | 2.1 ± 0.1 | 0.8 | - | - |
| 5 | TOC | % | 35.7 ± 1.9 | 39.9 | - | - |
| 6 | C/N | - | 17 ± 0.8 | 49.9 | 28.4 ± 0.4 | 25–35 |

Note: the value is demonstrated in the form of "Avg ± S.D".

### 3.4. Ratio Mixing

The amount of the base material, rice straw, was calculated in order to control the pH, moisture content and C/N ratio. The calculation is completed to achieve the optimal moisture content of 50–60%. With the average moisture of solid waste being 76.8%, 0.81 tonne of rice straw was mixed with 1.8 tonnes of solid waste (84.5% biodegradable solid waste) to achieve a moisture content of 55% (the average of the optimal range). As a result, the calculated ratio of solid waste to rice straw was 69:31.

The C/N ratio was checked again, and it was shown that the C/N ratio after mixing was 30.4, which is in the optimum C/N ratio (25–35, with the optimum value around 30) [24]. The pH of the mixed solid waste was 6.5, which is within the optimal pH range (6.5–7.5). Due to its low density, rice straw also helps to increase the porosity of the mixture. The mixture was applied for the composting process and divided into two windrows.

### 3.5. Composting Process

The compost windrows have been operational for 10 weeks, since 4 August 2022. The change in pH during composting process is illustrated in Figure 6. The pH was adjusted by mixing the solid waste and rice straw with an initial pH of 6.5. During the composting process, the pH fluctuated between 6.2 and 7.9. This pH range is optimal for the composting process [24]. The pH fluctuates more during the decomposition period than during the maturity period. This indicates that the composting process is stable during the maturity period. These findings could contribute to the temperature pattern, as discussed in the following section.

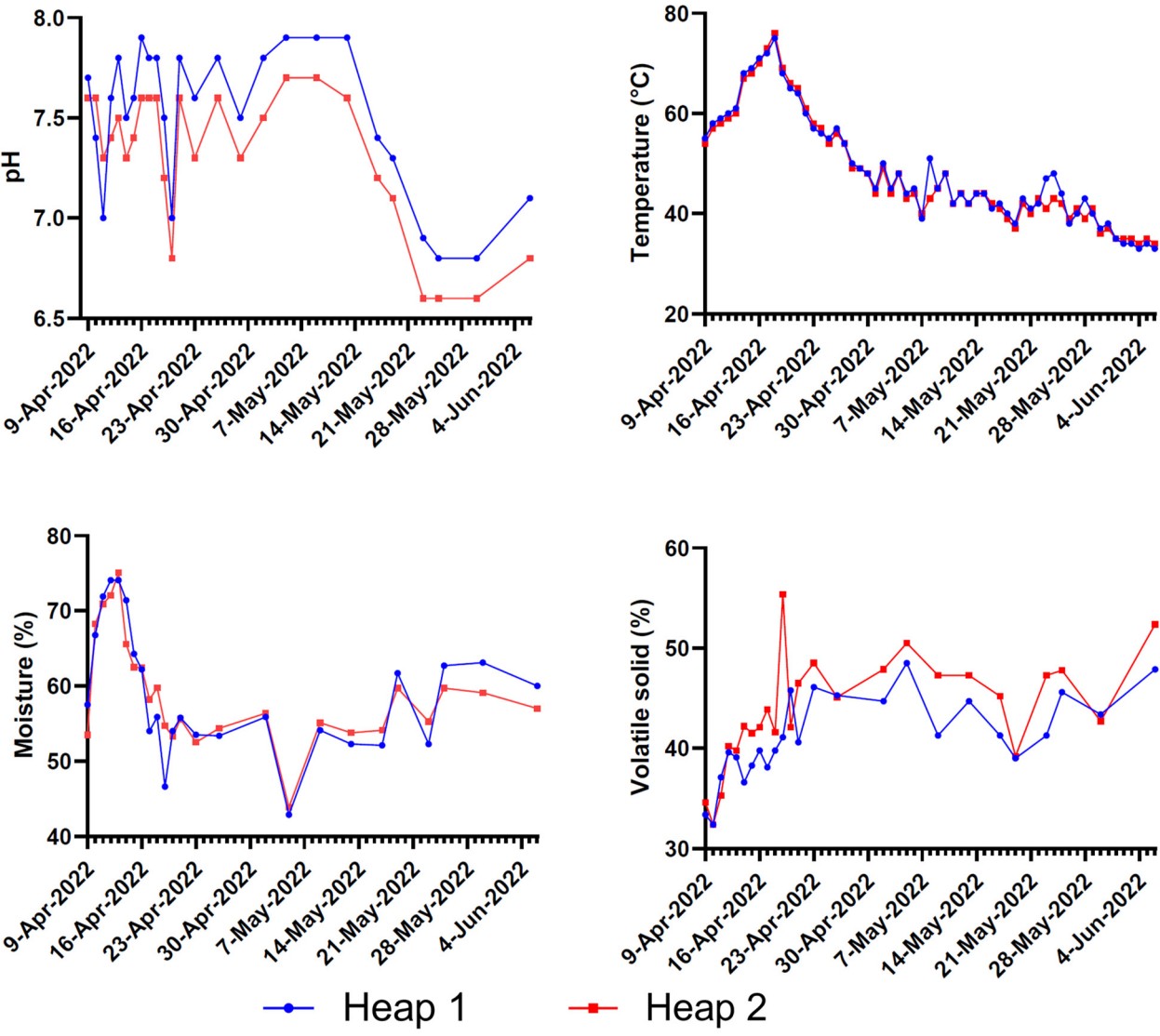

**Figure 6.** The fluctuation of pH, temperature, moisture content and volatile solid during the composting process.

Figure 6 depicts the temperature fluctuation during the composting process. The highest temperature of the two compost heaps was achieved on day 11. The highest temperatures of the two windrows were 75 °C and 76 °C, respectively. Within 15 days, the thermophilic phase (temperatures above 55 °C) was prolonged. The starting temperature of the reactors was found to be in the thermophilic phase on day 1 and quickly reached peak values within the first two weeks. This is due to the microorganisms breaking down the biodegradable organic matter and nitrogenous compounds [25–27]. The period of high temperature plays an important role in removing pathogens, parasite eggs, cysts and flies from the solid waste. The pathogen appearance were tested for the composting process's final product.

Previous research suggested that temperatures above 70 °C could be inhibitory to many microbes [28,29]. In this study, the windrow was manually mixed to release the heat of the windrow on the 11th day. This happened for only one day during the process. However, the useful microorganisms were tested for compost quality. If the windrow becomes too hot, turning or aerating will help to dissipate the heat.

The fluctuation of moisture content during the composting process is shown in Figure 6. The moisture content of the two heaps was under control. The mixtures of the solid waste

and rice straw of windrow 1 and windrow 2 were 57.5 and 53.5%, respectively. The moisture level rose over the next four days as a result of the mixture's leachate. The leachate was collected and re-added into the windrows. On day 6, the moisture content was reduced, and the optimum moisture was achieved on day 9. Aeration and manual mixing on a daily basis aided in the release of moisture content and kept it under control. Following this, the moisture content must be at least 50% [30].

　　　Figure 6 illustrates the change in volatile solid (VS), which indicated the reduction in organic matter. The VS was reduced from 41.1% to 32.4%. The result indicated that the organic matter was decomposed by the microorganism activity. This was shown by the change in temperature (Figure 6). The weight of the compost product was determined after 10 weeks using two windrows. The result showed that the products weighed 1.656 tonnes, corresponding to 36.5% organic waste lost through the composting process with 2.61 tonnes of ingredients (1.8 tonnes of organic waste and 0.81 tonne of rice straw). The organic waste loss ratio is slightly lower than in previous studies, ranging between 40 and 50% of the initial amount [31,32]. When the maturity period is extended, the weight of the compost product can be continuously reduced.

### 3.6. Quality of Compost Products

　　　After a 10-week composting process that included decomposition and maturation, the compost product is sampled and analysed in accordance with the requirements of Standard 10 TCN 526:2002. Four samples were taken and analysed from two windrows of compost. The result of the compost quality is shown in Table 2.

**Table 2.** The quality and standards of compost product. N.D indicates not detected; DW indicates dry weight and the value is demonstrated in the form of "Avg ± S.D".

| No. | Parameter | Unit | Compost Quality ($n = 4$) | Vietnam Standard 10TCN 526:2002 | United States Standard TMECC [33] | European Union Standard QP Compost [34] |
|---|---|---|---|---|---|---|
| 1 | pH | - | 7.6 ± 0.2 | 6.0–8.0 | 6.0–8.0 | 5.5–8.0 |
| 2 | Moisture | % | 52.5 ± 0.8 | <than 35% | 40–60% | 30–60% |
| 3 | T-N | % | 2.2 ± 0.1 | Not less than 2.5% | 1.0–3.0% | 1.0–4.0% |
| 4 | $P_2O_5$ | % | 1.2 ± 0.1 | Not less than 2.5% | 0.5–3.0% | 0.4–4.0% |
| 5 | $K_2O$ | % | 2.8 ± 0.1 | Not less than 1.5% | 0.5–3.0% | 0.2–3.5% |
| 6 | TOC | % | 26.6 ± 2.7 | Not less than 13% | - | - |
| 7 | Cu | mg/kg DW | 38.7 ± 6.6 | <200 mg/kg | <1500 mg/kg | 70–1000 mg/kg |
| 8 | Cd (LOD = 0.3) | mg/kg DW | N.D. | <2.5 mg/kg | <39 mg/kg | 0.7–10 mg/kg |
| 9 | Cr | mg/kg DW | 13.3 ± 1.6 | <than 200 mg/kg | <1200 mg/kg | 70–200 mg/kg |
| 10 | Ni | mg/kg DW | 7.6 ± 0.4 | <than 100 mg/kg | <420 mg/kg | 20–200 mg/kg |
| 11 | Pb | mg/kg DW | 12 ± 1.8 | <than 250 mg/kg | <300 mg/kg | 70–1000 mg |
| 12 | Zn | mg/kg DW | 135.5 ± 7.5 | <than 750 mg/kg | <2800 mg/kg | 210–4000 mg |
| 13 | Hg | mg/kg DW | 0.5 ± 0.03 | <than 2 mg/kg | <17 mg/kg | 0.7–10 mg/kg |
| 14 | Number of microorganisms | CFU/g | $1.2 ± 0.3 × 10^7$ | less than $10^6$ | No specific standard, but microbial activity is often measured | No specific standard, but microbial presence is expected |
| 15 | Salmonella | CFU/g | N.D | N.D | N.D | N.D |
| 16 | Particle size (pass through 2mm sieve) | % | 33.6 ± 3 | <4–5 mm | 2–5 mm | 2–5 mm |
| 17 | Particle size (pass through 5 mm sieve) | % | 100 | <4–5 mm | 2–5 mm | 2–5 mm |

According to the results, 13/16 of the compost's parameters met the standard. Cd and Salmonella, in particular, were not detected. Other heavy metal parameters were also significantly lower than standards. This result indicates that the compost could be used as a base fertiliser for planting. The nutrient content must also be checked before being applied to the plants.

The C/N ratio is a good indicator of the maturity level of an organic substance, as it significantly affects microbiological growth [29]. According to [35,36], the C/N ratio of mature compost should ideally be between 10 and 12. In this study, the C/N ratio is 12.1, which indicates a good degree of maturity.

In this study, the ingredient of the compost process is separated solid waste; therefore, the heavy metal parameters were lower than the requirements of the standards. This result could be explained by Tan An City's solid waste separation practice. Table 3 compares the heavy metal content of the compost product in this study and the previous study with the ingredient of municipal solid waste. The results show that the heavy metal concentration in the compost product was low because the solid waste was separated well and not contaminated by hazardous waste or chemical products. In terms of mercury, previous studies [37,38] found mercury concentrations in compost made from separated municipal solid waste at the source in the United States and Europe. The results show that the mercury concentration in this study is within the previous study's range. The presence of mercury in compost could be explained by the composition of Hg in fresh solid waste [38], whereas the amount of Hg in the ingredient was not determined in this study. According to [39], the mercury level in food waste ranges from 0.05 to 4.26 mg/kg of fresh solid waste.

**Table 3.** The comparison of heavy metal concentration in compost products in this study and previous studies.

| No. | Parameter | Unit | Compost Quality ($n = 4$) | Previous Studies |
|-----|-----------|------|---------------------------|------------------|
| 1 | Cu | mg/kg DM | $38.7 \pm 6.6$ | 150–300 * |
| 2 | Cd (LOD = 0.3) | mg/kg DM | N.D. | 2.00–4.00 * |
| 3 | Cr | mg/kg DM | $13.3 \pm 1.6$ | 80–100 * |
| 4 | Ni | mg/kg DM | $7.6 \pm 0.4$ | 40–50 * |
| 5 | Pb | mg/kg DM | $12 \pm 1.8$ | 250–350 * |
| 6 | Zn | mg/kg DM | $135.5 \pm 7.5$ | 300–655 * |
| 7 | Hg | mg/kg DM | $0.5 \pm 0.03$ | N.D.–0.63 ** 0.02–1.9 *** |

* [40]; ** [37]; *** [38].

In terms of compost stability, the test adhered to the guidelines of standard 10 TCN 526:2002. Every day, the temperature of a bag containing 10 kg of compost product was measured. The results showed that the temperature remained stable around 29 °C, indicating that the compost was stable.

As for nutrient content, the total nitrogen and phosphorus (indicate by $P_2O_5$) did not meet the fertiliser standards. Previous research suggested that pig or cow manure could be used as one of the raw material inputs for the composting process to improve the N and P contents in the compost product. In terms of moisture content, the moisture content of the compost product is higher than standard due to moisture control during the compost process. This matter is quite easy to deal with. During the maturation process, no more water is added into the heap. Currently, the maturity period is only two weeks; however, the period could be prolonged to more than 2 weeks (normally 4–6 weeks) to ensure the correct moisture content.

## 4. Discussion

### 4.1. Practical Implications

The Vietnamese Government has been actively engaged in establishing a comprehensive framework to facilitate the transition towards a circular economy. This commitment

is evident in the announcement and implementation of the LEP 2020, which explicitly acknowledges the importance of implementing a waste-circular economy. The LEP 2020 emphasises the significant role of solid waste in reducing the reliance on raw materials, extending product lifecycles, minimising waste generation and mitigating environmental harm. In line with these objectives, this study aims to showcase the viability of composting biodegradable organic MSW as a means of recycling and resource recovery.

Before implementing this project, WWF-Vietnam conducted several training and communication activities among households to increase their awareness as well as their practices in waste separation at the source. The outcome of this project has attracted more than 85% of Tan An residents to support and actively participate in waste separation at the source, thereby reducing 30%–40% of waste to be burned/deposed in landfills, contributing to raising people's awareness about solid waste management, saving resources, reducing waste treatment costs and highlighting the feasibility of the project [41].

Composting, a natural biological process, is critical in the circular economy because it effectively converts organic matter into nutrient-rich soil amendments. These composts have a wide range of applications in agriculture, landscaping and gardening, providing valuable nutrients to support plant growth and soil health [42]. Furthermore, composting provides additional environmental benefits, such as substantial reductions in greenhouse gas (GHG) emissions, the conservation of water resources and soil fertility enhancement. By integrating solid waste composting practices, the circularity of material flows is enhanced, contributing to the development of a more sustainable and resilient society.

*4.2. Theoretical Implications*

To initiate the circular economy approach, the study focuses on reducing waste generation and diverting waste away from landfills. In the case of Tan An City, it is estimated that approximately 47.2% of the municipal solid waste generated can be effectively recycled, leading to a significant reduction in landfill waste. This equates to a daily reduction of 61.4–70.8 tonnes or a yearly reduction of 22,396–25,842 tonnes. The composting technology being implemented not only caters to the biodegradable organic waste in Long An Province, but it also has the potential to be replicated in other regions of Vietnam. The results of the composting experiment conducted in the study area demonstrated that composting processes are feasible and effective for managing biodegradable organic waste.

In terms of nutrient recovery, the composting process implemented in Tan An City has the potential to recover 109.11–125.9 tonnes of nitrogen and 1854.93–2140.34 tonnes of carbon per year. These valuable carbon and nutrient sources are converted into soil amendments, enriching the soil with essential elements for agricultural activities. Given the importance of agriculture in Long An Province, with 350,000 hectares of agricultural land [43], diverting MSW from landfills and converting it into compost prevents the loss of valuable nutrients. This practice helps to mitigate the risk of environmental pollution, including the emission of landfill gases and the contamination of water and soil through leachate.

Managing greenhouse gas emissions is a concern within the waste industry. Previous studies have proposed GHG emission factors for the biodegradable organic fraction of MSW composting, with estimated values for methane, nitrous oxide and carbon dioxide emissions of $2.46 \times 10^{-2}$, $2.03 \times 10^{-2}$ and $5.63 \times 10^{-2}$ kg $CO_2$e (carbon dioxide equivalent) per kilogram of wet feedstock, respectively [44]. Based on these estimations, the total annual GHG emissions from the composting process in Tan An City ranges from 2266.5 to 2615.2 tonnes of $CO_2$e per year.

In comparison, estimating the GHG emissions associated with landfilling the same MSW requires referencing the IPCC 2006 guidelines for sanitary landfills, which suggest emissions of 0.72 tonne of $CO_2$e per tonne of landfilled waste. For Tan An City, this translates to approximately 16,125–18,606 tonnes of $CO_2$e emissions per year. These findings clearly show that composting biodegradable organic MSW leads to significantly lower GHG emissions compared to landfilling. Consequently, numerous previous studies advocate

for composting as a favourable alternative to landfilling, highlighting its recognition as an environmentally sound waste management approach in current research efforts.

## 5. Conclusions

The first sustainable Vietnamese separation-at-the-source MSW study was conducted in Tan An City, Long An Province, and the biodegradable solid waste content was studied for composting purposes. Based on the findings, it could be observed that the MSW separation programme in Vietnam has been progressing steadily, with only 0.4% of non-biodegradable waste being detected. This highlighted the potential of government activities to promote MSW separation at the source across the country.

The biodegradable solid waste collected during the study period contained 15.1% slow biodegradable organic matter, such as coconut shells, durian shells and tree branches, with the remainder being garden waste. The pH, moisture, VS, T-N, TOC and C/N ratio of biodegradable solid waste were $8.7 \pm 0.4$, $76.8 \pm 1.9\%$, $68.3 \pm 1.3\%$, $2.1 \pm 0.1\%$, $35.7 \pm 2.2\%$ and $17 \pm 0.8$, respectively. Rice straw was selected as the base material for composting, and its mixing ratio with biodegradable solid waste was 69:31 to achieve a moisture content of 55%. This is equivalent to 0.81 tonne of rice straw combined with 1.8 tonnes of solid waste (84.5% biodegradable solid waste). Within 10 weeks, including decomposition and maturation, the mixture was composted. It was found that the compost could be used for agricultural purposes, with only a small amount of nitrogen and phosphorus added to ensure sufficient nutrient levels for planting. Potential materials for enhancing the nutrient level of the compost include septage, manure, effluent wastewater from manure biogas digesters, etc. Composting contributes to lower environmental pollution costs associated with landfilling or, worse, open burning/dumping. As a result, sustainable composting with MSW separation at the source could pave the way for integrated MSW management in Vietnam.

**Author Contributions:** Conceptualization: T.L.H., T.K.O.L., Y.J.W. and T.L.T.; methodology: T.L.H., T.K.O.L. and T.L.T.; investigation: T.L.H., T.K.O.L. and C.T.P.; formal analysis: Y.J.W.; writing—original draft preparation: T.L.H. and Y.J.W.; writing—review and editing, T.K.O.L. and T.L.T.; visualization, T.L.H. and Y.J.W.; supervision, T.K.O.L.; project administration: T.K.O.L. All authors have read and agreed to the published version of the manuscript.

**Funding:** This research received no external funding.

**Data Availability Statement:** Data are available from the authors upon reasonable request.

**Acknowledgments:** The authors thank Van Lang university, Kyoto University of Advanced Science and WWF-VN for supporting this research. The authors would like to express their gratitude to WWF-Vietnam for allowing the use of the photos.

**Conflicts of Interest:** The authors declare that they have no known competing financial interests or personal relationships that could have appeared to influence the work reported in this paper.

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
