# Peer review of "Towards Sustainable Composting of Source-Separated Biodegradable Municipal Solid Waste—Insights from Long An Province, Vietnam"

_sustainability, doi:10.3390/su151713243_

Round 1
Reviewer 1 Report
The article describes the research of municipal solid waste sorting and biodegradable waste composting. Such results may be of interest to readers. However, to be accepted into the Journal, some corrections is needed. I think it can be accepted with minor revision. The comments are given below:
* Fig.1. The term "geological" is not suitable for this map.
* Chapter 2.2. It is necessary to write what amount of waste was shredded. What is the weight of each of the 5 samples?
* Figure 4. It is not clear why there is such a large amount of remaining waste among the separated MSW in the organic waste container. It is necessary to specify what kind of waste it is. And why they end up among biodegradable waste?
* Fig. 4. and further in the text. Use the term not "recycled" (they are not recycled yet), but "recyclable".
* Table 1. The characteristics of separated solid waste are presented. And where are the characteristics of rice straw?
* Table 2. Compost parameters compared to the requirements of the Vietnamese standard. For an international reader, it would be more interesting if there was a comparison with European or USA standards.
Author Response
Thank you for the comments and suggestions.
Please see the attachment for our point-by-point response.

Reviewer 2 Report
Overall Comment
“Towards sustainable composting of source-separated biodegradable municipal solid waste- Insights from Long An Province, Vietnam” is a meaningful project presented by the authors. However, as every research work presented, it needs a strong support on its novelty or originality. This point should be improved to strengthen the manuscript. The paper structure is also not very clear in this case and numerous formatting and typographical mistakes are spotted across the manuscript. Nonetheless, the paper in its current form is not publishable, and thus, the authors are required to address the comments before further consideration.
Abstract
From what I read, the work started with the characterisation of the MSW, thereby applying windrow composting process to produce the compost suitable for agricultural applications.
I recommend that the line 21 should be placed right after the problem statement, followed by the characterisation of MSW. What’s the testing used to validate the effectiveness of the compost? Please include the test results, especially numerical supports would be better.
Make sure past tense should be used in the abstract for the action point.
Introduction
1. The problem statement presented in line 66 – 72 is quite weak. Authors should also include studies / countries which have similar issues, and the ways of overcoming. Thereby, linking to the current study
2. “Circular economy” is not just a term to be simply used without actual application. Author should explain how this project is developed based on circular economy.
3. Author also did not fully explain how the feasibility study is conducted. Please include brief details.
Materials and Methods
1. Put the study period in section 2.1 as well
2. Section 2.2, line 110 & Section 2.3, line 153 should be placed in table form, along with the standard methods, like Table 2
3. Section 2.3 should be in past tenses, except for explanation
4. Composting study information and parameters are missing in this section. Rice straw etc was used as bulking materials. Each batch size? Temperature range?
Results and Discussion
1. Line 168 – 169 should be provided the basis and ref.
2. Line 173, how does the compost did not meet the fertiliser standards? Which parameters?
3. Figure 4 seemed to have look quality compared to other figures. Make sure all figures input are high quality image
4. Section 3.5, the writep Fig 3 and Fig 4 should be fig 6?
The paper structure is also not very clear in this case and numerous formatting and typographical mistakes are spotted across the manuscript.
Author Response

(The authors gave the same response as above.)

Reviewer 3 Report
The authors present study-case of well known process of municipal waste compost. The authors did not avoid mistakes:
Part 2.1:
In my opinion, the description Ward 3 of Tan An City is very general. The lack of details about characteristics of factors, which have impact on properties and quantity of municipal waste (for instance: living standards and consumption, size of bins, availability of bins, character of the development).
Conclusion:
I think that, the authors should showed proposition materials which will be used as raw materials of N and P.
References:
Part of the references is too old. e.g. 21, 24, 34, 35, 36.
Author Response

(The authors gave the same response as above.)

Reviewer 4 Report
This was a very interesting topic on the composting of source-separated biodegradable municipal solid waste in a Vietnam's province. The following parts must be revised prior to the publication of the article:
-Please provide the source(s) of the images displayed in Figures 1 and 3.
-You must briefly describe the methodology for each of the 17 parameters of determination. Just the provision of the compost quality standard number "10-TCN-448 526-2002" does not provide any information to the majority of the readers who do not have access to the respective protocols. You must mention at least in brief the methodologies to render the study reproducible.
-Use more critical comments in the Discussion section and also add more references to support your statements.
Minor editing of the English language required to improve the quality of the text.
Author Response

(The authors gave the same response as above.)
